# PET/CT in Patients with Breast Cancer Treated with Immunotherapy

**DOI:** 10.3390/cancers15092620

**Published:** 2023-05-05

**Authors:** Sofia C. Vaz, Stephanie L. Graff, Arlindo R. Ferreira, Márcio Debiasi, Lioe-Fee de Geus-Oei

**Affiliations:** 1Nuclear Medicine-Radiopharmacology, Champalimaud Center for the Unkown, Champalimaud Foundation, 1400-038 Lisbon, Portugal; 2Department of Radiology, Leiden University Medical Center, P.O. Box 9600-2300 RC Leiden, The Netherlands; l.f.de_geus-oei@lumc.nl; 3Division of Hematology/Oncology, Lifespan Cancer Institute, Providence, RI 02903, USA; sgraff1@lifespan.org; 4Legorreta Cancer Center, The Warren Alpert Medical School, Brown University, Providence, RI 02903, USA; 5Católica Medical School, Universidade Católica Portuguesa, 2635-631 Lisbon, Portugal; arr.ferreira@ucp.pt; 6Breast Cancer Unit, Champalimaud Center for the Unkown, Champalimaud Foundation, 1400-038 Lisbon, Portugal; marcio.debiasi@fundacaochampalimaud.pt; 7Biomedical Photonic Imaging Group, University of Twente, P.O. Box 217-7500 AE Enschede, The Netherlands; 8Department of radiation Science & Technology, Delft University of Technology, P.O. Postbus 5 2600 AA Delft, The Netherlands

**Keywords:** immunotherapy, immune checkpoint inhibitors, breast cancer, immuno-PET, PET/TC, FDG, biomarkers, mAb

## Abstract

**Simple Summary:**

Immunotherapy is beneficial in specific breast cancer types. To monitor the response to immunotherapy, positron emission tomography (PET) is useful. Several criteria to interpret these images are available and it is essential to have knowledge about their characteristics and applications. Furthermore, it is important to recognize imaging patterns of immune-related adverse events. In the metastatic setting in particular, immunotherapy is only administered if the tumor expresses receptors for the specific treatment, for which biopsies are required to determine receptor expression. However, due to the invasive character of the procedure, biopsies are generally not repeated over time and are not performed in multiple lesions. The strength of PET is that it permits whole-body imaging in a noninvasive way. Few studies in humans have been performed up to now. This narrative review summarizes the ongoing research on immunotherapy options for breast cancer and the role of (immuno-) PET in assessing therapy response.

**Abstract:**

Significant advances in breast cancer (BC) treatment have been made in the last decade, including the use of immunotherapy and, in particular, immune checkpoint inhibitors that have been shown to improve the survival of patients with triple negative BC. This narrative review summarizes the studies supporting the use of immunotherapy in BC. Furthermore, the usefulness of 2-deoxy-2-[^18^F]fluoro-D-glucose (2-[^18^F]FDG) positron emission/computerized tomography (PET/CT) to image the tumor heterogeneity and to assess treatment response is explored, including the different criteria to interpret 2-[^18^F]FDG PET/CT imaging. The concept of immuno-PET is also described, by explaining the advantages of mapping treatment targets with a non-invasive and whole-body tool. Several radiopharmaceuticals in the preclinical phase are referred too, and, considering their promising results, translation to human studies is needed to support their use in clinical practice. Overall, this is an evolving field in BC treatment, despite PET imaging developments, the future trends also include expanding immunotherapy to early-stage BC and using other biomarkers.

## 1. Introduction

Immunotherapy based on immune checkpoint inhibitors (ICI) has changed the therapeutic landscape in oncology by improving patient prognosis, including for those diagnosed with triple negative breast cancer (TNBC) [1,2]. However, optimal patient selection and response evaluation remain clinically challenging. Currently, to identify patients who may benefit from immunotherapy, a combination of immunohistochemical evaluation to confirm the absence of hormone and ErbB2 receptors (TNBC) is used in the early setting. In the metastatic setting, programmed-death ligand-1 (PD-L1) status, high microsatellite instability (MSI-high), and high tumor mutational burden (TMB-high) further contribute to select patients who may benefit from immunotherapy. Despite profound evaluation of the tumor and its microenvironment, other predictive biomarkers are not yet available [3].

In daily practice, positron emission tomography (PET) with 2-deoxy-2-[^18^F]fluoro-D-glucose (2-[^18^F]FDG) is routinely used in the diagnosis, staging, and monitoring (efficacy and safety) of patients with breast cancer (BC) receiving treatment, including for patients receiving immunotherapy. Despite its high sensitivity and clinical utility in the early identification of immune-related adverse events (irAEs)—such as inflammation in the thyroid, lungs, liver or colon—2-[^18^F]FDG PET is rather nonspecific; therefore, new radiopharmaceuticals that can identify specific immune system targets are under investigation in the preclinical and clinical settings. 

In this narrative review, we summarize (1) the pivotal trials supporting the use of immunotherapy in BC, (2) the association between 2-[^18^F]FDG uptake and tumor heterogeneity, (3) the role of 2-[^18^F]FDG positron emission/computerized tomography (PET/CT) in assessing the response to immunotherapy in patients with BC, and (4) the ongoing research in immuno-PET, specifically focusing on preclinical radiopharmaceuticals relevant for BC.

## 2. Tumor Microenvironment in the Intersection with Breast Cancer Immunotherapy

The tumor microenvironment (TME) is composed of infiltrating and resident host cells (among others, endothelial cells, fibroblasts, and immune cells) in association with secreted factors and extracellular matrix proteins that together modulate tumor growth [4]. In BC, both tumor-infiltrating lymphocytes (TILs) and gene expression profiling (GEP) signatures related to the TME were shown to incorporate prognostic and predictive value, thus supporting the active relationship between tumors and TME [5]. Overall, the TME quality seems to embody the highest prognostic relevance in HR (hormone receptors)-/HER2- and HER2+ tumors, which contrasts with HR+/HER2- tumors for whom tumor proliferation is the strongest parameter predicting clinical outcomes [6]. The types of cells and their spatial distribution provide additional information about the implications of TME quality in shaping disease behavior [7,8]. Recognizing such value, the canonical definition of the intrinsic subtypes of BC is evolving from a tumor-cell-centered categorization to incorporating TME-related information, especially immune-related information [9,10,11]. As a result, novel disease entities with implications on expected response to immunotherapy are now considered, namely immune desert (“cold” tumors), margin-restricted, stroma-restricted, and fully inflamed tumors [11]. This information is, however, indissociable from the dynamic nature of tumors, for example, increased tumor mutational burden and reduced major histocompatibility complex (MHC) class I, and the host’s immune status (e.g., immunosuppressive action of certain chemotherapies) and their relationship. Hence, TME features co-evolve with the tumor and the host’s immune system over time, namely from primary tumor location to metastasis (as across metastases) and after exposure to therapy [12,13]. The incorporation of these elements into therapeutic development and disease monitoring is crucial to increase the impact of immunotherapy.

## 3. Clinical Selection of Immunotherapy for Breast Cancer

The development of ICIs in BC treatment took longer compared to in other tumors, due to their limited immunogenicity. Currently, the use of immunotherapy is the standard of care in TNBC, a heterogeneous group of tumors which stand out as the more genomically unstable and, consequently, the more immunogenic BC subtype. 

Pembrolizumab is a humanized monoclonal antibody (mAb) which targets the programed-death-1 receptor (PD-1) [14]. The KEYNOTE-355 study tested pembrolizumab associated with chemotherapy (paclitaxel, nab-paclitaxel, or the combination of carboplatin and gemcitabine) against chemotherapy with a placebo for patients with metastatic disease [15]. Patients were excluded if evidence of active autoimmune disease, immunosuppressive therapy, or active brain metastases were present. Comprehensive systemic staging to document the extent of metastatic disease included either a combination of CT scans of the chest, abdomen, and pelvis with contrast plus a bone scan, or 2-[^18^F]FDG-PE/CT, the latter adding value when equivocal findings were reported on other imaging modalities [1]. Pembrolizumab proved to improve overall survival (OS) in the subgroup of patients whose tumors expressed PD-L1 (Combined Positive Score [CPS] ≥ 10) with a median OS of 23.0 vs. 16.1 months (hazard ratio [HR] 0.73; 95%CI 0.55–0.95; *p* < 0.01) compared to chemotherapy alone. Based on the KEYNOTE-355 study, both the Food and Drug Administration (FDA) [16] and European Medicines Agency (EMA) [17] have granted approval to pembrolizumab, in combination with chemotherapy, for treating patients with metastatic TNBC (mTNBC) expressing PD-L1 (CPS ≥ 10). Pembrolizumab is also approved by the FDA for tumors with MSI-high, mismatch repair proficiency, or TMB-high. This represents the agency’s first tumor-agnostic approval supported by the results of the KEYNOTE-158 trial [18]. Although it is an important advance in BC treatment, caution is necessary concerning its clinical use, due to the limited number of BC patients included in the trial and the low frequency of MSI-high in BC (0–1.5%) [19]. 

Atezolizumab is another ICI approved by the EMA to treat mTNBC. It is a humanized monoclonal antibody that targets the PD-L1. The IMpassion130 trial evaluated the addition of atezolizumab to nab-paclitaxel in first line mTNBC. Despite showing a statically significant progression-free survival (PFS) benefit in the intention-to-treat (ITT) population (7.2 vs. 5.5 months, HR 0.80; 95%CI 0.69–0.92; *p* < 0.01), the study could not be considered positive because it failed to achieve a statistically significant impact on its co-primary endpoint of OS in the ITT population (21.3 vs. 17.6 months; HR 0.84; 95%CI 0.69–1.02; *p* = 0.08) [20]. In the PD-L1 positive subgroup analysis, however, it seemed that OS was significantly improved (25.0 vs. 15.5 months; HR 0.62; 95%CI 0.45–0.86), but no formal statistical comparison could be conducted as the hierarchical analysis planned for the study had already been considered negative for the co-primary outcome. In the subsequent IMpassion131 trial, atezolizumab was combined with paclitaxel instead of nab-paclitaxel as the chemotherapy backbone but failed to replicate the findings of IMpassion130 [21]. This led to the voluntary withdrawal by the drug sponsor of the initial FDA conditional approval [22]. In contrast, the EMA approval remains active.

Even though ICIs seem to work only in the PD-1/PD-L1-positive subpopulations in the metastatic setting, in early stage or curative BC, benefit is not restricted to the biomarker-selected population. PD-L1, which is not present in normal breast tissue, is present in about half of BC, with increasing prevalence in high-grade and high-proliferation BC such as TNBC [23,24]. Translational research has shown that the major cytotoxic chemotherapies, including those included in KEYNOTE-522, upregulate PD-L1 expression and promote the anti-tumor immune response [25]. Given that pembrolizumab relies on an active immune system to exert an anti-tumor effect, the goal of treatment is to increase antigen presentation with the cytotoxic effects of chemotherapy, upregulate PD-L1 expression, and ultimately promote an ICI-mediated anti-tumor effect. Therefore, in the neoadjuvant early-stage BC (EBC) setting, patients need not be selected for ICI based on PD-L1 expression. The KEYNOTE-522 trial included patients with clinical T1c N1-N2 or T2-4 N0-2 stage untreated TNBC, and randomized them to receive pembrolizumab or a placebo, together with standard neoadjuvant chemotherapy up to one year after surgery. Overall pathological complete response (pCR) rate was raised from 51.2% to 64.8% (*p* < 0.01), regardless of PD-L1 status. Long-term outcomes have also been improved by pembrolizumab with a 3-year event-free survival (EFS) rate of 85% vs. 77% (HR 0.63; 95%CI 0.48–0.82; *p* < 0.01) [26,27]. These results granted FDA and EMA approval for pembrolizumab in the treatment of high-risk early-stage TNBC [17,28]. 

In the same EBC setting, atezolizumab has shown to improve pCR in the IMpassion031 trial (41%1 to 57.6%, *p* < 0.01), but the long-term outcomes of the GeparDouze/NSABP B-59 trial are still awaited before its use can be better assessed in the early setting (NCT02008227) [29]. GeparNuevo also studied ICI with durvalumab in EBC, and although the addition increased pCR numerically, it did not reach statistical significance. However, the study’s unique design allowed an immune priming phase with some patients receiving a single dose of ICI two weeks prior to neoadjuvant chemotherapy. Among those who received the priming dose, there was a significantly higher pCR [30]. The main clinical trials about ICIs in TNBC previously mentioned in the text are summarized in Table 1.

These early-stage approvals are based on clinical staging, which incorporates a combination of clinical examination and imaging, including breast and whole-body systemic imaging. Guidelines recommend systemic imaging where concern for metastases exist, noting that 2-[^18^F]FDG PET/CT may add further benefit in resolving equivocal findings on conventional imaging modalities and identifying unsuspected regional nodal disease and/or metastatic disease [1,26,27]. Patients are excluded if they have autoimmune disease requiring systemic treatment or a history of immunodeficiency, given the unique mechanism and adverse event profile of ICIs.

## 4. Future Trends in Immunotherapy

The future of immunotherapy will include expanding indications beyond TNBC, enhancing patient selection, escalation/de-escalation strategies to optimize the dosing and length of ICI exposure, and mitigation of irAEs. 

Combination ICI and patient selection criteria for ICIs were explored in the Phase 2 NIMBUS trial, which enrolled patients with HER2-negative advanced breast cancer (TNBC and hormone-receptor positive [HR+]) and TMB-high rather than utilizing PD-L1 expression [31]. Patients received dual-ICI with nivolumab Q14 days and ipilimumab Q6 weeks, and were followed for ORR. ORR was observed in 13.3% (4/31) of patients, and notably 3 patients remained progression-free for at least 15 months. PD-L1 and HR were not predictive of ORR, while TMB ≥ 14 yielded increased ORR. This is particularly important given that PD-L1 did not distinguish responders and non-responders in the neoadjuvant setting [26]. I-SPY2, an adaptive phase 2 trial, also included patients with HR+ EBC treated with pembrolizumab, improved pCR from 13% to 30% [32]. These trials laid the groundwork to explore alternative ICI patient selection criteria and expand treatment to HR+ BC. The differential response seen in GeparNuevo, with the use of durvalumab priming, mentioned above, also creates opportunity for improved ICI sensitivity or expanded patient selection [30]. 

Beyond TMB, other biomarkers of interest in predicting BC ICI response include TILs, gains in CD274, and expression of MHC-II. TILs may predict favorable disease independent of treatment and MHC-II was identified retrospectively to be predictive in I-SPY2 [33,34]. In an exploratory analysis in patients with advanced breast cancer treated with durvalumab, HR for ICI efficacy with CD274 gain/amplification was 0.18 (95%CI = 0.05–0.71, *p* = 0.0059), with CD274 normal/loss ICI efficacy HR = 1.12 (95%CI = 0.42–2.99, *p* = 0.8139) [35]. 

In the final EFS analysis, KEYNOTE-522 showed similar three-year EFS among patients who achieved pCR, regardless of whether they received pembrolizumab or placebo, which raised the question of whether the one year of adjuvant pembrolizumab created additional benefit or not, particularly given similar outcomes with GeparNeuvo which lacked the one year of adjuvant pembrolizumab [36,37]. Furthermore, imaging strategies to predict pCR, particularly early in planned neoadjuvant treatment, could drive innovative trial design. Likewise, the worse outcome of patients without pCR, despite the five-drug chemotherapy-ICI combination in KEYNOTE-522, begs for novel approaches. Adding therapies already approved in high-risk EBC, such as olaparib or capecitabine, or adding ICIs alone (clinical trials NCT02954874 and NCT02926196) or in combination with antibody drug conjugates (ADCs—clinical trial NCT04434040) each represents potential opportunities to improve outcomes [38,39,40], which again may be better selected or predicted by earlier imaging prediction of response or radiomic assessments for tumor heterogeneity. Combinations of ADC and ICI are being explored in advanced breast cancer, across tumor subtypes as well (clinical trials NCT05382286, NCT04468061, NCT04448886 and NCT04538742). irAEs, particularly skin toxicities and endocrinopathies, are prevalent among patients treated with ICI and, to date, there are no clear predictors of toxicity to identify patients at the highest risk. I-CHECKIT, a prospective observational study, hopes to shed light on which patients are at the highest risk for irAEs. Radiomics or theranostics could play a role in monitoring ICI toxicity and improving safety profile. 

## 5. Correlation between 2-[^18^F]FDG Uptake and Tumor Heterogeneity

Some studies have documented intra-individual heterogeneity between primary tumors and metastases, leading to the recommendation to repeat biopsies during the course of the disease to assess, e.g., PD-1/PD-L1 status. Considering that some tumors are inaccessible for biopsy and it is impossible to biopsy the complete tumor lesion to assess heterogeneity, PET/CT may address these needs following a non-invasive approach. PET/CT evaluates the whole-body expression of specific receptors using targeted radiopharmaceuticals that can be used for diagnosis, staging, and to guide individualized treatment. A retrospective study from Xie et al. demonstrated that in patients with mTNBC, treated with first-line immunotherapy, the median PFS was significantly longer in patients with low-intratumoral heterogeneity (IATH) on 2-[^18^F]FDG PET/CT than in patients with high-IATH (9.4 vs. 5.8, HR = 0.3, 95%, CI 0.1–0.8, *p* = 0.022) [41]. Multivariate analysis demonstrated intertumoral heterogeneity (IETH) as an independent predictor of PFS (9.4 vs. 4.9 months, HR = 0.3, 95%, CI 0.1–0.7, *p* = 0.01) and, in the survival receiver operator characteristics (ROC) analysis, IETH also showed the highest area under the curve (AUC) of 0.69 [41]. Consequently, baseline IETH, derived from 2-[^18^F]FDG PET/CT, could represent a simple and promising predictor of disease outcomes.

In patients with BC, it has been reported that highly heterogeneous tumors tend to be associated with less immune cell infiltration, less activation of the immune response, and worse overall survival, thereby influencing the immunotherapy outcome [41,42]. Hiraka et al. hypothesized that 2-[^18^F]FDG uptake reflects tumors’ aggressive features and inflammation, which represent BC microenvironment [43]. They found significant associations between SUVmax and the degree of TILs (r = 0.428, *p* < 0.001) and between SUVmax and PD-L1 positivity (r = 0.413, *p* < 0.001) [43]. Furthermore, Fuji et al. observed a significant correlation between 2-[^18^F]FDG uptake and neutrophil/lymphocyte ratio (NLR) (r = 0.323, *p* < 0.001), which is an indicator of systemic inflammation, and noticed that patients with either low SUVmax or low NLR presented no recurrent disease [44]. The same group verified that platelet/lymphocyte ratio (PLR) (r = 0.376, *p* < 0.001) is also independently associated with SUVmax, but not with recurrent disease [45]. Although additional research is necessary to determine whether 2-[^18^F]FDG PET/CT can be used as an immunological biomarker to select candidates who might benefit from immunotherapy in patients with BC, these examples demonstrate the usefulness of considering 2-[^18^F]FDG uptake as a potential prognostic marker.

Furthermore, tumor SUVmax seems to be a multifaceted biomarker. On the one hand, early-stage TNBC patients with high tumor SUVmax on pre-treatment 2-[^18^F]FDG PET/CT are more likely to display pCR after neoadjuvant therapy. On the other hand, high tumor SUVmax acts as an unfavorable prognostic factor, since this pattern is associated with higher rates of recurrence after treatment.

In contrast to the immunohistochemical analysis, 2-[^18^F]FDG PET/CT may be a more objective tool through quantitative analysis of lesion’s uptake, thus reducing unconformities among different exams or physicians.

## 6. 2-[^18^F]FDG PET/CT Interpretation & Immunotherapy Response Assessment

To monitor treatment response to ICIs using 2-[^18^F]FDG PET/CT, a baseline evaluation of 2-[^18^F]FDG-avid lesions is mandatory, followed by interim evaluations, commonly every 8–12 weeks after the start of treatment. 2-[^18^F]FDG PET/CTs, at other time-points, should be performed in case of suspicion of progression or clinical deterioration (Figure 1 and Figure 2). Furthermore, 2-[^18^F]FDG PET/CTs should be performed before discontinuation of immunotherapy, and before restarting immunotherapy to reestablish a new baseline for subsequent treatment evaluation [46].

Response monitoring of patients treated with ICIs is often challenging and not as straightforward as with conventional cytotoxic or targeted anticancer treatments. Beyond the conventional tumor response patterns, there are four specific immunotherapy response patterns which need to be considered: pseudoprogressive disease, dissociated response, hyperprogressive disease, and durable response [46]. Pseudoprogressive disease is defined as an increase in size of already existing tumor lesions, or the appearance of new lesions followed by tumor regression. This can occur up to several months after immunotherapy in circa 10% of patients, but most frequently starts within the first 4–6 weeks of treatment. This phenomenon could result from several mechanisms, including local edema due to inflammation, delayed immune response, or immune cell infiltration in the tumor lesions [47]. Another hypothesis is the delayed efficacy of ICIs. In around 10% of patients, another phenomenon is observed: dissociated response or mixed response or disproportional response. It usually presents as the decrease or stabilization of some tumor lesions, with a concomitant increase in other lesions [48]. It is important to discriminate dissociated response from homogeneous progression, since patients with a dissociated response could benefit from the continuation of immunotherapy, in combination with local therapy (surgery, radiotherapy, or local ablative therapy) of the oligoprogressive lesions [49]. Hyperprogressive disease occurs in 4–29% of patients and is defined as a more than twofold increase in tumor growth rate during immunotherapy or treatment failure within 2 months of start of treatment, leading to premature death [50,51]. The underlying mechanism is not yet clearly understood, despite active investigation [46]. In contrast, there are also patients who will achieve a durable response that can persist for several years, even after stopping ICIs. A durable response is observed more often with ICI therapy than with chemotherapy/targeted therapies (25% vs. 11%) [52].

Since pseudoprogression and hyperprogression cannot be discriminated on the first evaluation, a “wait-and-see” strategy has been recommended. In this scenario, reevaluations every 4–8 weeks to prevent patients without clinical deterioration from prematurely terminating immunotherapy is recommended by the iPERCIST criteria [53] (Table 2). Following the iPERCIST criteria, patients with progressive metabolic disease—according to PET Response Criteria In Solid Tumors (PERCIST 1.0) [54]—at the first 2-[^18^F]FDG PET/CT evaluation, are classified as having unconfirmed progressive metabolic disease (uPMD). In such cases, it is recommended to repeat 2-[^18^F]FDG PET/CT after four to eight weeks, to establish confirmed progressive metabolic disease (cPMD). Approximately 30% of these patients are revealed to have pseudoprogressive disease and dissociated response, and will benefit from continuing ICIs. Similar initiatives to iPERCIST include the “PET Response Evaluation Criteria for IMmunoTherapy” (PERCIMT) [55], the “immunotherapy-modified PERCIST, five-lesion analysis” (imPERCIST5) [56] and the “PET/CT Criteria for Early Prediction of Response to Immune checkpoint inhibitor Therapy” (PECRIT) [57]. These criteria, in contrast to iPERCIST, do not require follow-up PET/CT to confirm disease progression [57] (Table 2).

Additionally, when interpreting a 2-[^18^F]FDG PET/CT examination, it is important to assess immune organs and to determine whether the spleen is enlarged and/or shows increased uptake, or an increased spleen-to-liver ratio (SLR), or bone marrow-to-liver ratio (BLR). Furthermore, it is important to identify potential irAE, of which the most common are hypophysitis, pneumonitis, colitis, hepatitis, and thyroiditis [46]. This may enable an intervention in life-threatening cases, and will help changing therapy in less severe cases [58].

Incorporation of these metabolic response criteria into future prospective randomized trials will be crucial for both validation and to understand their impact on long-term patient outcome. Additional steps to understand 2-[^18^F]FDG PET/CT interpretation in the setting of tumor heterogeneity will also need to be undertaken.

## 7. Immuno-PET

Immuno-PET consists of mAb molecular imaging using PET, obtained through the intravenous administration of a mAb radiopharmaceutical. It enables the combination of the high-sensitivity and quantitative capabilities of PET, with the specificity and selectivity of a mAb for a tumor cell-surface marker [59]. It plays a potential role in cancer immunotherapy, by providing three-dimensional, whole-body, non-invasive, tumor biomarker expression cartography. Therefore, it provides identification of malignant lesions in patients and information about inter- and intra-tumoral heterogeneity, as well as, the dynamics of cancer response to treatment [2,47,48]. By guiding the optimal use of FDA-approved mAbs and complementing the information obtained from tissue analyses, immuno-PET may also reduce the risks associated with invasive procedures (such as biopsy and surgery) and save healthcare costs, impacting patients’ quality of life. 

Although immuno-PET is not yet approved in clinical practice, it has already been used in several cancer subtypes with promising results. The main clinical studies on immuno-PET in solid cancers are summarized in Table 3, where different radiopharmaceuticals are mentioned, depending on the targeting molecule and type of cancer. Niemeijer et al. analyzed immuno-PET in patients with non-small cell lung cancer (NSCLC) and verified that tumor PD-L1 and PD-1 expression was accurately determined non-invasively using [^18^F]BMS-986192 and [^89^Zr]Nivolumab PET/CT, respectively [60]. They demonstrated that immuno-PET was able to assess tumor heterogeneity and that radiopharmaceutical uptake correlated with PD-L1 expression and tumor-infiltrating immune cells on immunohistochemistry [60]. Furthermore, Farwell et al. developed an anti-CD8 radiolabeled minibody, named [^89^Zr]Df-IAB22M2C, to determine tumor CD8+ leukocyte distribution in patients with metastatic solid cancers (melanoma, NSCLC and hepatocellular carcinoma were included) [61]. The authors reported a safe procedure and a good tumor-to-background ratio when images were acquired 24 h post-injection; therefore, they concluded that this may enable the early prediction of response to immunotherapy [61]. The phase II clinical trial (NCT03802123—iCorrelate) was completed in November 2022 and its results will provide stronger evidence about this immuno-PET radiopharmaceutical.

Although immunotherapy in BC is an emerging field, in the last decade, some clinical data have been shared, showing promising results for imaging specific immune system targets in patients with BC. 

Currently, the most common PET targets in patients with BC include HER (Trastuzumab and Pertuzumab) and vascular endothelial growth factor (VEGF-A—Bevacizumab) [62,63,64,65,66,67,68]. Few studies evaluated anti-CEA [59,69]. Only one clinical study, Bensch et al., targeted PD-L1 (Atezolizumab) [70]. All studies were prospective and some included clinical trials. The average number of included patients was 23 (range: 6–56). Zirconium-89 was the radioisotope used to label anti-HER, anti-VEGF and anti-PD-L1 mAbs. Cupper-64 and Gallium-68 were used to label anti-HER and anti-CEA mAbs, respectively. Overall, the following protocol was used by the different groups: (1) intravenous administration of the cold mAbs, (2) intravenous administration of the radiopharmaceutical (around 37 Mq), and (3) early and delayed imaging acquisition. The best imaging time differed among the studies, but Dijkers et al. [62] suggested 4–5 days post-injection as being the best time period for imaging acquisition after ^89^Zr-trastuzumab injection. No significant adverse effects were reported, only grade I and grade II allergic reactions after ^89^Zr-trastuzumab administration in two patients [65].

All studies reported good tumor-to-background ratios and mentioned that pre-treatment with the cold mAb improved the ratio. Most of them concluded that the radiopharmaceutical was useful to detect tumors. This year, it was demonstrated how tumor uptake could be quantified in immuno-PET. A recent study evaluated ^89^Zr-anti-EGFR tumor uptake in 10 patients with wildtype K-RAS colorectal cancer and ^89^Zr-anti-HER3 uptake in 5 HER3-positive solid tumors [71]. The authors observed that both Patlak linearization (which evaluates distribution volume and net influx rate values, representing reversible and irreversible uptake, respectively), and SUV and tumor-to-plasma/tumor-to-blood ratio (TPR/TBR) measured at late time points (5 to 6 days post-radiopharmaceutical injection) with constant mass administered doses, enabled the quantification of irreversible tumor uptake [71]. These results may be useful to assess the response to treatment quantitatively with immuno-PET.

As previously mentioned, several clinical studies have already demonstrated the feasibility of [^89^Zr]trastuzumab, but only one clinical study targeting PD-L1 (Atezolizumab) shared their results [70]. In 2018, Bensch et al. published the results from 22 patients with locally advanced or metastatic bladder cancer, NSCLC and TNBC, and reported that liver metastases presented the highest uptake and lung metastases the lowest. They also verified that TNBC metastases presented a lower uptake (average was SUVmax = 6.4) compared to those from the bladder and NSCLC (average was SUVmax = 12.8 and 10.5, respectively) and found that the tumor objective response rate (ORR) was 25% for TNBC (compared to 56% for bladder cancer and 11% for NSCLC) [70]. The role of pre-treatment [^89^Zr]atezolizumab PET/CT is being investigated in clinical trials (NCT02478099 and NCT02453984), the results of which are expected to improve patient selection for ICIs.

However, several results from preclinical studies have been published showing the potential of radiopharmaceuticals targeting anti-PD-L1 in BC cell lines and animal models [72,73,74,75].

Several other preclinical studies demonstrated the use of PET radiopharmaceuticals in labelling clusters of differentiation, such as antihuman syndecan-1 (CD138) mAb labelled with ^124^I or ^131^I [76]; YY146 (CD146) mAb labelled with ^52^Mn, ^89^Zr or ^64^Cu [77,78]; and CD8+ T cells labelled with ^64^Cu or ^89^Zr [79,80]. Few studies published their results on CTL4-A (CD152) labelling with ^64^Cu, and demonstrated good visualization of cytotoxic T-lymphocyte antigen-4 (CTLA-4) on the T cell infiltrating tumor [81,82]. CTLA-4 expression in tumors was assessed by PET imaging acquired 48 h post-injection (region-of-interest was used for semiquantitative analysis), ex vivo biodistribution studies, and tissue staining, confirming tissues’ CTLA-4 positivity.
cancers-15-02620-t003_Table 3Table 3Main clinical studies about immuno-PET in solid cancers.Targeting MoleculeRadiopharmaceuticalCancer**HER-2**[^89^Zr]trastuzumab[^89^Zr]pertuzumabBreast cancer [62,65,66][^64^Cu]DOTA-trastuzumabBreast cancer [67,68]**PD-1**[^89^Zr]pembrolizumab [^89^Zr]nivolumab NSCLC [60,83]Melanoma [83]**PD-L1**[^89^Zr]atezolizumab Metastatic bladder cancer, NSCLC, and TNBC [70] [^18^F]BMS-986192 NSCLC [60]**CD8**[^89^Zr]Df-IAB22M2CMelanoma, lung cancer, and hepatocellular carcinoma [61,84]**EGFR or VEGF**[^89^Zr]cetuximab [^89^Zr]bevacizumab NSCLC [85,86] Head and neck [86]Breast cancer [63,64]**CEA and HSG**[^68^Ga]IMP288 Breast cancer [59,69]Colorectal cancer [87]Medullary thyroid carcinoma [88]**PSMA**[^89^Zr]IAB2MProstate cancer [89,90]**CA-IX**[^89^Zr]girentuximabRenal cell carcinoma [91]**CA 19-9**[^89^Zr]HuMab-5B1NCT02687230 (ongoing)Pancreatic ductal adenocarcinoma**CTLA-4**[^89^Zr]ipilimumab (NCT03313323 ongoing)Melanoma**MUC16 and CD3**[^89^Zr]REGN4018 (NCT03564340 ongoing)Ovarian cancer**CA 19-9**—cancer antigen 19-9; **CA_IX**—carbonic anhydrase IX; **CD8** – cluster of differentiation 8; **CEA**—carcinoembryonic antigen; **CTLA-4**—cytotoxic T-lymphocyte associated protein 4; **EGFR**—epidermal growth factor receptor; **HER-2** – human epithelial growth factor receptor 2; **HSG**—Histamine-Succinyl-Glycine; **MUC16**— mucin 16 (CA125); **NSCLC**—non-small cell lung cancer; **PD-1**—programmed death receptor 1; **PD-L1**—programmed death ligand 1; **PSMA**—prostate specific membrane antigen; **TNBC**—triple negative breast cancer; **VEGF**—vascular endothelial growth factor.

## 8. Future Molecular Imaging Developments and New Radiopharmaceuticals Pipeline

The concept of targeting the same molecule (or mAb) with different radioisotopes for diagnostic or therapeutic purposes, is termed theranostics. Its main idea is to treat what we see. Hence, besides using anti-PD-L1 mAbs for diagnosis, if labelled with therapeutic isotopes, they can be used for therapy, and it has started to be explored in BC models using ^225^Ac-DOTA-labeled anti-PD-L1 antibodies [92]. 

Fibroblast activation protein inhibitors (FAPIs) have shown promising results in the diagnosis of several types of cancer, because of their enhanced tumor-to-background ratio, including in in BC. FAPIs may be labelled with both imaging (^18^F- or ^68^Ga-FAPI) and therapeutic (^177^Lu- or ^90^Y-FAPI) radioisotopes. Several papers have reported better diagnostic accuracy of FAPIs compared to 2-[^18^F]FDG PET/CT in BC [93,94]. Moreover, some case reports demonstrated the clinical usefulness of treating BC with FAPI [95,96]. Some groups are developing preclinical models to explore the combination of immunotherapy and radionuclide therapy targeting FAP, trying to increase the therapeutic efficacy mediated by the abscopal effect [97]. Other preclinical studies are trying to image FAP expression on activated fibroblasts of the tumor stroma, with the aim of predicting and monitoring therapeutic response to FAP-targeted CAR T cell therapy [98].

Radiomics and artificial intelligence is an active field of research. Some groups are analysing how tumor metabolic heterogeneity, intensity, shape and texture features could be used as biomarkers to predict tumor biology and to select the best candidates for a specific therapy. Radiomics consists of extracting a large number of quantitative parameters from medical images, based on the hypothesis that such features could be linked to genotypic and molecular characteristics of the tumor lesions. An undoubted advantage of radiomics is the possibility of following all lesions over time in a non-invasive way [99,100]. Many studies have demonstrated the potential role of 2-[^18^F]FDG PET/CT in different clinical settings of BC [100]. The next step is to confirm and to standardize this evidence in validation studies using big data, before these approaches can be used for precision medicine being implemented in daily clinical practice.

Granzyme B is a serine protease downstream effector of cytotoxic T cells and a useful predictive biomarker for a good response to immunotherapy. Granzyme B based-PET (GZP-PET) imaging provides three-dimensional information on effector cell activation, allowing for the noninvasive identification of the TME that can be useful in monitoring the immune response over time [101].

The sodium iodide symporter (NIS) was proposed as a non-immunogenic radionuclide reporter in ErbB T1E28z chimeric antigen receptor (CAR) therapy in two TNBC models: MDA-MB-231 with high PD-L1 expression, and MDA-MB-436 with low PD-L1 expression. Considering that PD-L1 expression correlates inversely with CAR-T tumor retention, the human NIS was used to quantify tumor retention of pan-ErbB family targeted CAR-T by PET, ex vivo and non-invasively. The authors verified that CAR-T tumor retention was inversely correlated with immune checkpoint expression in TNBC models. This is an interesting example of CAR-T imaging through PET [102].

Recently, Sriraman et al. shared the results of their pioneering research on ^18^F-labeled anti-human CD8 VHH (small antibody against human CD8) in mice with leukaemia cell line xenografts, which enabled the acquisition of high-quality immuno-PET imaging one hour after the radiopharmaceutical intravenous injection, in contrast to ^89^Zr-labelled tracers which have an optimal imaging quality several days after injection only [103]. The ^18^F-labelled tracers improve patient comfort, reduce radiation exposure, and facilitate the logistics and schedules of the nuclear medicine department [103].

## 9. Conclusions

Immunotherapy with ICIs is an expanding standard of care in the clinical management of patients with BC. Currently, 2-[^18^F]FDG PET/TC plays an important role in the monitoring of the response to immunotherapy and the detection of irAEs. Several interpretation criteria are available, but guidance to uniformize PET/CT interpretation and quantification is still lacking.

Although many preclinical studies on immuno-PET in BC are available in the literature, only one clinical study on labelled PD-L1 has been published. The preliminary results are promising, with good tumor-to-background ratios and no significant adverse effects being reported. Hence, immuno-PET seems to be a potential tool for in vivo mapping the immune biomarkers in both spatial and temporal distribution, non-invasively and systematically. However, results from clinical trials are awaited.

Combining the information on specific biomarkers, obtained by both micro- (immunohistochemistry) and macroscopic (immuno-PET) evaluations, will certainly increase the knowledge about the tumor and TME behavior, which will be useful to better understand immunotherapy in the clinical practice.

## Figures and Tables

**Figure 1 cancers-15-02620-f001:**
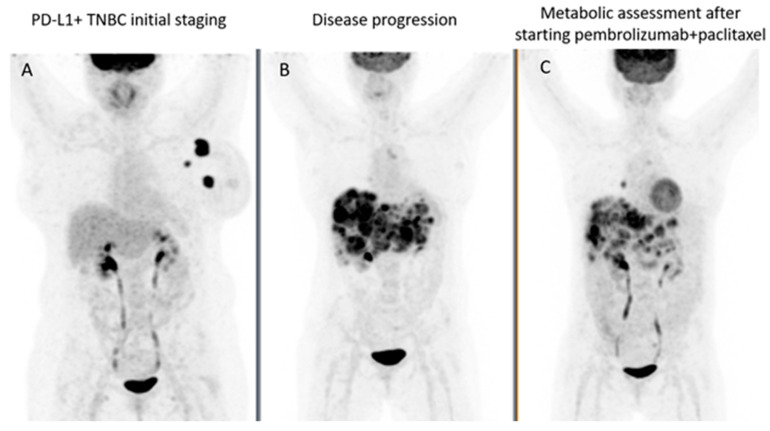
(**A**) Maximum intensity projection images on initial staging 2-[^18^F]FDG PET/CT of a 63 year-old woman with locally advanced TNBC, T2N + M0, Ki67 90% confirmed left breast primary cancer with ipsilateral axillary lymph nodes involvement. She was submitted to neoadjuvant chemotherapy followed by mastectomy with axillary dissection, radiation therapy and maintenance chemotherapy. Recurrence was suspected four years later; (**B**) 2-[^18^F]FDG PET/CT revealed extensive liver involvement, abdominal lymph nodes, lung metastases only seen on CT component, and unifocal bone metastasis in L2. Para-gastric lymph node histology, obtained via upper endoscopy, confirmed TNBC PD-L1 positive. Pembrolizumab and paclitaxel were initiated, leading to partial metabolic response on follow-up 2-[^18^F]FDG PET/CT performed 4 months later, as shown in; (**C**) liver and lymph nodes partial metabolic response, and complete metabolic response in the bone lesion—more details are given in Figure 2. Mediastinal and right pulmonary hilar lymph nodes increased 2-[^18^F]FDG uptake could be due to reactive inflammatory process, probably an irAE. *2-[^18^F]FDG PET/CT investigations performed at the Nuclear Medicine-Radiopharmacology, Champalimaud Foundation, Lisbon, Portugal*.

**Figure 2 cancers-15-02620-f002:**
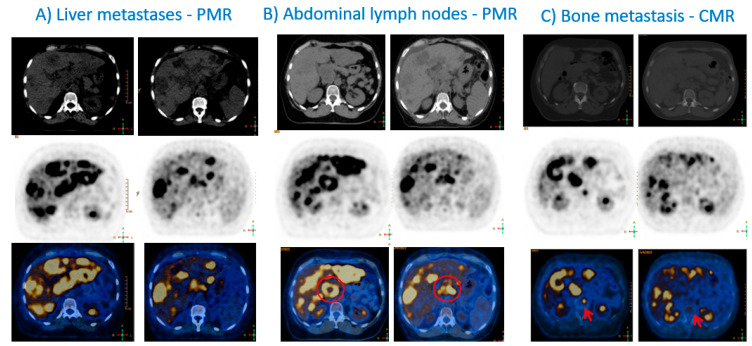
This figure displays axial sections of CT component (top row), 2-[^18^F]FDG (middle row) and fused PET/CT (bottom row) before (left for each column) and after starting pembrolizumab and paclitaxel (right for each column). Partial metabolic response (PMR) in liver (**A**) and abdominal lymph nodes (red circle) (**B**). Metastases was achieved at the end of months of combined pembrolizumab and paclitaxel therapy. Complete metabolic response (CMR) in the L2 bone lesion was obtained (red arrow in (**C**)). *2-[^18^F]FDG PET/CT investigations performed at the Nuclear Medicine-Radiopharmacology, Champalimaud Foundation, Lisbon, Portugal*.

**Table 1 cancers-15-02620-t001:** Main pivotal trials of Immune Checkpoints Inhibitors in Triple-Negative Breast Cancer.

Trial	Setting	Drugs	Nº of Patients	Primary Outcomes	Main Results	Comments
**KEYNOTE-355** [15] Phase III, randomized, placebo-controlled	aTNBC, first line	Pembrolizumab/Placebo + chemotherapy (paclitaxel, nab-paclitaxel, or carbo-gemcitabina)	847	Co-primary efficacy endpoints: PFS and OS assessed in the CPS ≥ 10, CPS ≥ f, and ITT populations	PFS (PD-L1 positive: CPS ≥ 10): 9.7 vs. 5.6 months (HR 0.65, 95% CI 0.49–0.86; one-sided *p* = 0·0012). OS (PD-L1 positive: CPS ≥ 10): 23.0 vs. 16.1 months (HR 0.73; 95%CI 0.55–0.95; *p* < 0.01).	Based on this trial, FDA and EMA have granted Pembrolizumab approval for first-line aTNBC in PD-L1 positive subpopulation (CPS ≥ 10).
**KEYNOTE-158** [18] Phase II, open-label one-arm trial	aBC, second or plus lines	Pembrolizumab monotherapy	233 (5 BC)	Objective response rate per RECIST in confirmed MSI-H/dMMR advanced noncolorectal cancer who experienced failure with prior therapy	Objective response rate: 34.3% (95% CI, 28.3% to 40.8%). Median PFS: 4.1 months (95% CI, 2.4 to 4.9 months). Median OS: 23.5 months (95% CI, 13.5 months to not reached).	FDA first tumor-agnostic approval.
**Impassion130** [20] Phase III, randomized, placebo-controlled	aTNBC, first line	Atezolizumab/Placebo + nab-paclitaxel	902	Co- primary efficacy endpoints: PFS (in ITT and PD-L1–positive subgroups tested with SP142 assay) and OS (tested in ITT—if positive, than tested in the PD-L1–positive subgroup)	PFS (ITT): 7.2 vs. 5.5 months (HR 0.80; 95%CI 0.69–0.92; *p* < 0.01) PFS (PD-L1 positive: SP142): 7.5 vs. 5.0 months (HR 0.62; 95%CI 0.49–0.78; *p* < 0.01) OS (ITT): 21.3 vs. 17.6 months (HR 0.84; 95%CI 0.69–1.02; *p* = 0.08) OS (PD-L1 positive: SP142): 25.0 vs. 15.5 months (HR 0.62; 95%CI 0.45–0.86). Not formally tested due to the hierarchical statistical analysis plan comparison could be conducted as the hierarchical analysis planned for the study had already been considered negative for the co-primary outcome. In the subsequent	EMA granted definitive approval for Atezolizumab based on Impassion130. FDA, however, granted partial approval that was withdrawn after the negative results of the confirmatory trial Impassion131.
**Impassion131** [21] Phase III, randomized, placebo-controlled	aTNBC, first line	Atezolizumab/Placebo + paclitaxel	651	PFS tested hierarchically first in the PD-L1-positive (SP142 assay) population and then in the ITT population.	PFS (PD-L1 positive: SP142): 6.0 vs. 5.7 months (HR 0.82, 95%CI 0.60–1.12; *p* = 0.20) PFS (ITT): 5.7 vs. 5.6 months (HR 0.86, 95%CI 0.70–1.05; not tested)	After these negative results, the label was dropped in the US
**KEYNOTE-522** [26] Phase III, randomized, placebo-controlled	eTNBC	Neoadjuvant Pembrolizumab/Placebo + chemotherapy (carbo-paclitaxel followed by AC/EC) followed by adjuvant Pembrolizumab/Placebo	1174	pCR and EFS in ITT	pCR: 64.8% vs. 51.2% (∆ 13.6%; 95%CI, 5.4–21.8%; *p* < 0.00) EFS: median not-reached. HR 0.63, 95%CI 0.43–0.93; *p* not reported)	After this study, pembrolizumab has been approved for neo + adjuvant use in high-risk eTNBC by the FDA and EMA
**Impassion031** [29] Phase III, randomized, placebo-controlled (only in the neoadjuvant phase)	eTNBC	Neoadjuvant Atezolizumab/Placebo + chemotherapy (nab-paclitaxel followed by AC) followed by adjuvant Atezolizumab (in the experimental arm)	455	pCR in ITT and PD-L1 positive populations	pCR (ITT): 58% vs. 41% (∆ 17%; 95%CI 6–27%; one-sided *p* = 0·0044) pCR (PD-L1 positive: SP142): 69% vs. 49% (∆ 20%; 95%CI 4–35%; one-sided *p* = 0·021)	Despite these positive results, atezolizumab was not approved for the neo + adjuvant treatment of eTNBC due to the lack of proof of long- term benefit
**GeparDouze/NSABP B-59** (NCT02008227) Phase III, randomized, placebo-controlled	eTNBC	Neoadjuvant Atezolizumab/Placebo + chemotherapy (carbo-paclitaxel followed by AC/EC) followed by adjuvant Atezolizumab/Placebo	1550	Co-primary: EFS and pCR	Not yet published	If positive, Atezolizumab shall receive approval for eTNBC
**GeparNuevo** [30] Phase II, randomized, placebo-controlled	eTNBC	Durvalumab/Placebo + chemotherapy (nab-paclitaxel followed by EC)	174	pCR	pCR: 53.4% vs. 44.2% (∆ 9.2%; *p* = 0.287).	Durvalumab effect was seen only in the window cohort (pCR 61.0% versus 41.4%)

**AC:** doxorubicin + cyclophosphamide; **aTNBC**: advanced Triple-Negative Breast Cancer; **aBC:** Breast Cancer; **EC:** epirubicin + cyclophosphamide; **EFS:** Event-Free Survival; **EMA:** European Medicines Agency; **eTNBC:** early Triple-Negative Breast Cancer; **FDA:** Food and Drug Administration; **HR:** Hazard-Ratio; **ITT:** Intention-to-Treat; **MMR:** Mismatch Repair; **MSI:** Microsatellite Instability; **OS**: overall survival; **pCR:** pathological complete response at the time of definitive surgery; **PFS:** Progression-Free Survival; **pts:** patients; **RECIST:** Response Evaluation Criteria in Solid Tumors.

**Table 2 cancers-15-02620-t002:** Summarized characteristics of the different 2-[^18^F]FDG PET/CT criteria to assess response to immunotherapy in solid cancers.

Criteria	PECRIT 2017 [57]	PERCIMT 2018 [55]	iPERCIST 2019 [53]	imPERCIST5 2019 [56]
Time for confirming PMD	3–4 weeks	3 months	2 months	3 months
Target lesions	minSUL * = 1.5 × mean SUL liver	Size (metabolically active lesion) > 1.0 or 1.5 cm	minSUL * = 1.5 × mean SUL liver	minSUL * = 1.5 × mean SUL liver
New lesions	Progression	PMD	iuPMD	Include in the sum of SULpeak, PMD if SULpeak > 30%
**Complete metabolic** **response (CMR)**	Disappearance of all target lesions			
**Partial metabolic** **response (PMR)**	↓≥30% from baseline	Disappearance of some metabolically active lesions without new lesions	↓ SULpeak in target lesions ≥ 30%	↓ sum of SULpeak in target lesions ≥ 30% and absolute ↓ SUL units ≥ 0.8
**Stable metabolic disease (SMD)**	Neither of the other options apply			
**Progressive metabolic** **disease (PMD)**	↑≥20% in the nadir of the sum of target lesions (>5 mm)	≥4 new lesions of <1 cm or ≥3 new lesions of >1 cm or ≥2 new lesions of >1.5 cm	iuPMD *: ↑≥30% in SULpeak or new metabolically active lesions PMD: PET confirmation 4–8 weeks after	↑>30% in SULpeak, with ↑SUL unit > 0.8

Table adapted from the “Joint EANM/SNMMI/ANZSNM practice guidelines/procedure standards on recommended use of [^18^F]FDG PET/CT imaging during immunomodulatory treatments in patients with solid tumors version 1.0” [57]. All these criteria consider ≤ 5 target lesions per patient. * **iuPMD**—immune unconfirmed metabolic progressive disease; **minSUL**: minimum SUL value in the tumor.

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
