# Peer review of "PET/CT in Patients with Breast Cancer Treated with Immunotherapy"

_cancers, 2023, doi:10.3390/cancers15092620_

Round 1
Reviewer 1 Report
This is an interesting review summarizing the ongoing research on immunotherapy options for breast cancer and the role of (immuno-)PET in assessing therapy response.
The topic is highly relevant and such review is perfectly justified.
However, I have several comments:
- In the abstract, you say that “Immunotherapy is only administered if the tumour expresses receptors for the specific treatment, for which biopsies are required to determine receptor expression”. This is correct for advanced/metastatic stages but not for early-stage in the neoadjuvant setting. Indeed, TNBC patients could be treated with neoadjuvant chemotherapy plus pembrolizumab regardless if the PD-L1 expression. Please clarify.
- Please remove the % when using CPS values (e.g. line 118).
- Tumor SUVmax seems to be a multi-faceted biomarker. On one hand, early-stage TNBC patients with high tumor SUVmax on pre-treatment FDG-PET imaging are more likely to display pCR after neoadjuvant therapy. On the other hand, high tumor SUVmax acts as an unfavorable prognostic factor since this pattern is associated with higher rates of recurrence after treatment. This should be stated.
- One hypothesis related to pseudo-progression is the delayed efficacy of immune checkpoint inhibitors. This should added.
- You didn’t mention 18F- or 68Ga- FAPI as a potential novel radiotracer. Can you please add a paragraph on it?
Author Response
àREVIEWER 1
This is an interesting review summarizing the ongoing research on immunotherapy options for breast cancer and the role of (immuno-)PET in assessing therapy response.
The topic is highly relevant and such review is perfectly justified.
However, I have several comments:
- In the abstract, you say that “Immunotherapy is only administered if the tumour expresses receptors for the specific treatment, for which biopsies are required to determine receptor expression”. This is correct for advanced/metastatic stages but not for early-stage in the neoadjuvant setting. Indeed, TNBC patients could be treated with neoadjuvant chemotherapy plus pembrolizumab regardless if the PD-L1 expression. Please clarify.
Thank you for your relevant comment. We corrected the sentence by specifying the metastatic setting, as follows: “In the metastatic setting in particular, immunotherapy is only administered if the tumour expresses receptors for the specific treatment, for which biopsies are required to determine receptor expression.”
- Please remove the % when using CPS values (e.g. line 118).
Done accordingly.
- Tumor SUVmax seems to be a multi-faceted biomarker. On one hand, early-stage TNBC patients with high tumor SUVmax on pre-treatment FDG-PET imaging are more likely to display pCR after neoadjuvant therapy. On the other hand, high tumor SUVmax acts as an unfavorable prognostic factor since this pattern is associated with higher rates of recurrence after treatment. This should be stated.
Thank you for this input. A paragraph was added in the end of the new section nº 5, as follows: “Furthermore, tumor SUVmax seems to be a multi-faceted biomarker. On one hand, early-stage TNBC patients with high tumor SUVmax on pre-treatment 2-[18F]FDG PET/CT are more likely to display pCR after neoadjuvant therapy. On the other hand, high tumor SUVmax acts as an unfavorable prognostic factor since this pattern is associated with higher rates of recurrence after treatment”.
- One hypothesis related to pseudo-progression is the delayed efficacy of immune checkpoint inhibitors. This should added.
We appreciate this remark and we added it in section nº 6, as follows: “This phenomenon could result from several mechanisms, including local edema due to inflammation, delayed immune response or immune cells infiltration in the tumour lesions [48]. Another hypothesis is the delayed efficacy of ICIs. In around 10% of patients, another phenomenon is observed: dissociated response or mixed response or disproportional response. It usually presents decrease or stabilization of some tumour lesions with a concomitant increase of other lesions [49].”
- You didn’t mention 18F- or 68Ga- FAPI as a potential novel radiotracer. Can you please add a paragraph on it?
We agree it is relevant to mention FAPI as a potential new radiopharmaceutical and a paragraph about FAPI was included in new section nº8 (lines 456-466).
Reviewer 2 Report
This narrative review paper summarizes on the one hand the current knowledge and perspectives of immunotherapy in breast cancer, and on the other hand the position and perspectives of PET imaging in immunotherapy. Although it is not a systematic review, the article is very interesting and provides useful insight in both reviewed fields.
The authors should address the following points:
- Section 3: The manuscript would benefit from a table summarizing the main trials of immunotherapy in BC.
- Section 4: pg4, ln186: Please specify that this is a FDG PET study.
- Section 5: pg6, ln257-260: The authors should note hat these criteria – in contrary to iPERCIST- do not require a follow-up PET/CT for confirmation of progression.
- Section 6: Please also mention the immunoPET studies by Niemeijer et al (2018) and Farwell et al (2022).
- Section 6: I would suggest the authors to create a table summarizing the main clinical immunoPET trials, not only necessarily in BC.
- Section 8: I think that this section would better fit earlier in the manuscript, e.g. directly after Section 3.
Author Response
àREVIEWER 2
This narrative review paper summarizes on the one hand the current knowledge and perspectives of immunotherapy in breast cancer, and on the other hand the position and perspectives of PET imaging in immunotherapy. Although it is not a systematic review, the article is very interesting and provides useful insight in both reviewed fields.
The authors should address the following points:
- Section 3: The manuscript would benefit from a table summarizing the main trials of immunotherapy in BC.
A table summarizing the main trials on immunotherapy in BC was added in section nº 3 – “Table 1. Main pivotal trials of ICIs in Triple-Negative Breast Cancer”.
- Section 4: pg4, ln186: Please specify that this is a FDG PET study.
Done accordingly in line 238.
- Section 5: pg6, ln257-260: The authors should note that these criteria – in contrary to iPERCIST- do not require a follow-up PET/CT for confirmation of progression.
This information was added in line 314 and 315.
- Section 6: Please also mention the immunoPET studies by Niemeijer et al (2018) and Farwell et al (2022).
In new section 7 about immuno-PET a paragraph about immuno-PET in other solid cancers was included (lines 370-386), mentioning the studies from Niemeijer et al and Farwell et al, as suggested.
- Section 6: I would suggest the authors to create a table summarizing the main clinical immunoPET trials, not only necessarily in BC.
We found this comment very interesting and we made a literature search to select the more relevant clinical studies about immuno-PET. In table 3 we summarized the main clinical studies and referred to several different radiopharmaceuticals according to the targeting molecule and type of cancer.
- Section 8: I think that this section would better fit earlier in the manuscript, e.g. directly after Section 3.
We agree with the suggestions of the reviewer and changed the text accordingly
Round 2
Reviewer 1 Report
The revised manuscript is of higher quality, and the changes made by the authors have significantly improved the work's clarity, rigour, and relevance. I believe that their work will make a valuable contribution to the field. As a result, the manuscript is now acceptable for publication.